# Green Commercial Aviation Supply Chain—A European Path to Environmental Sustainability

**Batoul Modarress Fathi** [1], **Al Ansari** [2,*] and **Alexander Ansari** [1]

1    Quality Technology Inc., Edmonds, WA 98026, USA
2    Department of Marketing, Seattle University, Seattle, WA 98122, USA
*    Correspondence: aansari@seattleu.edu

**Abstract:** The last century has witnessed European commercial aviation flourishing at the cost of environmental degradation by boosting greenhouse gas and $CO_2$ emissions in the atmosphere. However, the outcry for net-zero emissions compels the sector's supply chain to a minimum 55% reduction of greenhouse gas emissions below the 1990 level by 2030 and zero $CO_2$ emissions by 2050. This study examines a European environmental sustainability path toward a green commercial aviation supply chain. Driven by literature and a review of related documents, two propositions were advanced to orient perspectives on the relationship between pollution and the commercial aviation supply chain and actions being taken toward environmental sustainability. In semi-structured interviews, seventeen aerospace associates endorsed pollution sources in the commercial aviation supply chain during the four stages of the aircraft life cycle, including extracting the raw materials, manufacturing, ground and flight operations, and end-of-service. They recommended transitioning into green commercial aviation through the widespread deployment of innovative technologies, from modifying airframes to changing aviation fuel, utilizing alternative propulsion systems, adopting circular manufacturing, and improving air traffic management.

**Keywords:** aerospace; environmental pollution; greenhouse gas emissions; climate change

## 1. Introduction

Commercial aviation has become indispensable by contributing to the European region's mobility, connectivity, and economy. Between 2005 and pre-pandemic 2020, with revenue streams of EUR 52.1 billion, the sector grew at an average annual rate of 5%, adding 4.1% to the European GDP. Driven by increased regional and international flights, longer distances, the use of larger aircraft, and a record load factor of 83.3%, passenger kilometers escalated by 90%. The sector was supported by a 12 million workforce, while passenger flights increased from 2.1 billion to 4.6 billion in 2020. Moreover, the cargo volume from the European Union (EU27) and the European Free Trade Association (EFTA) increased by 60% [1].

With the COVID-19 outbreak, international and domestic flights dropped by 75.6% and 56%, respectively. However, cargo transport increased by 21.0%. The recovery year was 2022, as passenger flights increased to 86% of the pre-pandemic level [2,3]. The increasing number of airlines and aircraft in service supported the rising flights. In 2022, the world fleet size was 28,674 aircraft, including 6845 airplanes in Europe, of which 23,513 were actively operated by over 5000 airlines. Thus, demand for jet fuel escalated, leading to rising greenhouse gas and carbon dioxide ($CO_2$) emissions.

Between 1990 and 2019, the world's commercial aviation $CO_2$ emissions increased by an average of 203% per year due to an increase in global jet fuel consumption from 68 billion gallons to 95 billion gallons, generating 21.1 pounds of $CO_2$ per gallon burnt [4]. $CO_2$ emissions dropped from 1000 million tons in 2019 to 600 million in 2020 due to the COVID-19 outbreak. Air traffic emissions totaled 720 million tons in 2021, regaining nearly one-third of the fall in 2020 [5].

Still, according to the legally binding international United Nations treaty adopted by 196 countries, the immediate target is to reduce greenhouse gas emissions by 55% from 2010 levels by 2030 compared to the 1990 level and zero $CO_2$ emissions by 2050 [5]. The goal is to constrain global warming at less than 1.5 degrees Celsius of the pre-industrial levels. Thus, as signatories, European nations and their commercial aviation must curb greenhouse and $CO_2$ emissions [6].

Further concerns are perceived aircraft noise by commercial flights that exposed 3.6 million people across Europe in 2017. The noise jeopardizes people's well-being, especially in communities near airports, experiencing annoyance, sleep disturbance, ear barotrauma, cardiovascular diseases, premature mortality, and youth learning impairments [7,8]. More concerns are related to waste generated in flights, airports, commercial aviation services, and various stages of the aircraft life cycle [9].

This paper examines a European environmental sustainability path toward a green supply chain in commercial aviation in two steps. The first is to review the sources of pollution in European commercial aviation. The second is to validate commercial aviation pollution and obtain recommendations for transitioning into green supply chain commercial aviation as a path to environmental sustainability. This two-stage study is organized into five sections. First is a literature review on the relationship between aviation and environmental pollution during the four stages of the aircraft life cycle, including extracting the raw materials, manufacturing, ground and flight operations, and end-of-service. The second section describes a list of targets for a green commercial aviation supply chain. The third section presents a methodology to gain experts' perspectives on commercial aviation air pollution and a path toward green supply chain commercial aviation. The fourth section is the qualitative assessment of perceptions of European commercial aviation's environmental degradation. The fifth section is the participants' perspectives, followed by the conclusion in section six.

## 2. Literature Review

### 2.1. Relationship between Aviation and Environmental Pollution

According to the European Environmental Agency, climate change is anthropogenically coupled with the Earth's natural phenomena [10]. Trexler and Pincetl stated that humans had changed the Earth's natural systems so fast that they have driven the Earth's geology and ecosystems into the Anthropocene Epoch, leading to anthropogenic radiative forcings. The Intergovernmental Panel on Climate Change associate climate change with anthropogenic radiative forcings, which refers to atmospheric concentrations of greenhouse gases, particularly its 78% carbon dioxide ($CO_2$) component, affecting the Earth's energy balance over time [11–13]. The $CO_2$ accumulates and acts as a heating source [14]. Ibarrarán et al. [15] linked the anthropogenic radiative forcing to the frequency, intensity, and longevity of weather-related disasters. Gao et al. [16] related radiative forcing to wildfires, earthquakes, ocean raising, permafrost, increasing wetlands, mudslides, volcano eruptions, loss of global biodiversity, and species extinction.

Between 1990 and 2015, global net emissions of greenhouse gases increased by 43%, while $CO_2$ emissions increased by 51%. Consequently, the accumulated heat reached its highest level in the last 800,000 years by the second decade of the 21st century. In 2022, the amount of $CO_2$ in the Earth's atmosphere reached 419 ppm, warmed oceans to more than 337 zettajoules, accelerated decays in Arctic ice by 12.6% per decade since 1979, decreased ice sheets by 447 billion metric tons per year, and increased sea levels by four inches since 1998 [17].

The Intergovernmental Panel on Climate Change propagated climate change based on scientific observations and mathematical modeling and concluded that humans have heated planet Earth beyond recovery [13]. The most observed indicators of climate change caused by aviation are all greenhouse gases, namely, $CO_2$ and non-$CO_2$ species, including methane-$CH_4$, nitrogen oxide-$N_2O$, hydrofluorocarbons (HFCs), perfluorocarbons (PFCs), sulfur hexafluoride ($SF_6$), and nitrogen trifluoride ($NF_3$). Furthermore, pollutants are carbon monoxide (CO), sulfur dioxide ($SO_2$), nitrous oxides (NOx), ammonia (NH3), black carbon

(BC), organic carbon (OC), and non-methane volatile organic compounds (NMVOCs). CO increases concentrations of harmful atmospheric ozone ($O_3$) and vapor trails (contrails) coupled with BC formed by partially burning fossil fuels, biofuel, and biomass. These are the particles with naturally occurring, soot-causing human morbidity and premature mortality [18]. They also are the causes of rapid and extensive biodiversity loss, human health issues, and climate change [19,20].

Duxbury and Goodess et al. stated that human activities expedite climate change by burning fossil fuels that generate greenhouse gas emissions. In 2019, the European Environmental Agency also associated greenhouse gas emissions to waste (3%), agriculture (10%), industrial processes (9%), fuel combustion, fugitive discharges from fuel (53%), and transportation (25%), leading to the Earth's energy balance—either a warming or cooling effect over time, signaling urgency for mitigation [10,20,21]. According to the United Nations Environment Program, the 106 billion gallons commercial jet fuel market in 2020 is projected to grow to over 230 billion gallons by 2050, doubling greenhouse gas emissions [22].

Mitigating the climate forcing of air transport represents a formidable challenge, especially given the rapid demand growth and its contribution to European economic growth. Special attention is given to the global aviation sector, including domestic and international passenger and freight. This sector accounts for 1.9% of greenhouse gas emissions, 2.5% of $CO_2$ emissions, and 3.5% of effective radiative forcing impacting climate change. Historical observation by the European Aviation Environmental Report revealed that although greenhouse gas emissions decreased by 31.6% between 1990 and 2020, the transportation sector's share increased from 14.8% to 25.8% [23]. Land transport was the largest source of greenhouse gas emissions in the transportation sector, followed by aviation with 13.9%. The International Civil Aviation Organization predicted that international aviation greenhouse gas emissions will triple the 2015 level by 2050 and significantly impact climate change unless emissions are reduced by 90% (compared to 1990 levels) [1].

### 2.2. Pollution during Aircraft Lifecycle

Pollution accumulates throughout an aircraft's life cycle from extracting raw materials (stage 1), aircraft manufacturing (stage 2), flight operations (stage 3), and end-of-services (stage 4), which underlines the significance of resource overconsumption and environmental degradation (Figure 1) [20,23].

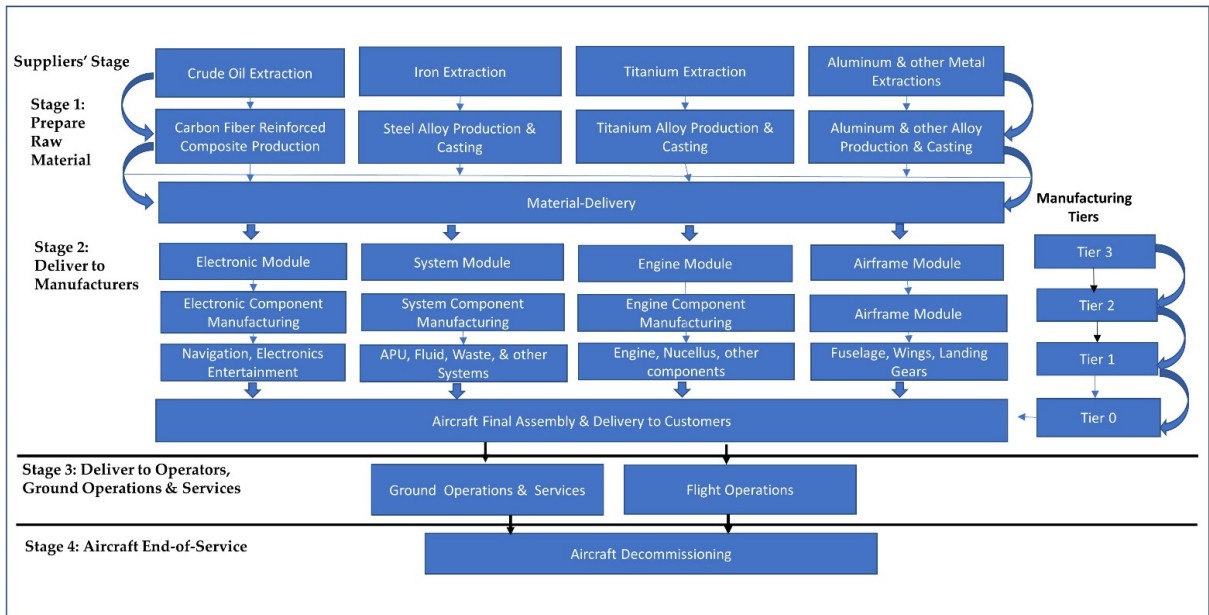

**Figure 1.** Aircraft life cycle from extracting raw materials to manufacturing & the end-of-service [20,23].

### 2.2.1. Stage One—Prepare Raw Materials

Aircraft supply chain manufacturing uses a variety of metal components, such as titanium, nickel, and aluminum. Other materials for electric and electronic components are lithium for advanced energy storage, nickel, graphite, cobalt, neodymium, and dysprosium for the permanent magnet in highly efficient electric motors, and yttrium for superconducting electric components. Besides metals, other materials include a host of plastics parts in aircraft, wooden materials used for maintenance and interior pieces, rubber for sheeting, tubing, adhesive-backed rubber, and general maintenance, paper for general maintenance, and fabrics for seats and interior. Extracting the raw material and processing leave a large quantity of residual waste, such as rocks and mill tailings, in addition to creating environmental impacts, such as soil degradation, water acidification, biodiversity loss, damage to ecosystem functions, and worsening climate change. This process generates approximately 1.8 billion tons of mineral processing waste yearly in the United States [24], while Europe generates 503.8 million metric tons [23].

### 2.2.2. Stage Two—Manufacturing

The aviation supply chain continues with aircraft manufacturing parts and components for final product assembly that requires various materials weighing more than 90%, from aluminum alloys, titanium alloys, steel alloys, and carbon-fiber-reinforced polymer composites [25]. According to the United Nations Environmental Program, resource extraction has tripled since 1970, including a five-fold increase in non-metallic minerals and a 45% increase in fossil fuel use [22]. The global aviation industry is the main customer of mined metals and other natural resources. At the same time, mining and processing raw materials, fossil fuels, and agriculture contribute to 50% of the world's greenhouse gas emissions and over 90% to the loss of biodiversity and water pollution. By 2060, global materials use could double the 2019 level, while greenhouse gas emissions could increase by 43% [23].

In manufacturing, unwanted materials, such as dust and particles during the manufacturing and transportation time, contaminate air and water. Additionally, chemical pollution, for instance, vapor, gases, moisture, and hazardous (hazardous organic solvents for degreasing) materials are used, leaving wastes that pose significant threats to human health and the environment. In aerospace manufacturing, over three hundred waste streams have been identified that are classified as solid and liquid. Hazardous solid wastes include composites, aluminum, titanium, steel alloys, and foam; liquids include paint, oil, and petroleum products. Other typical wastes generated are halogenated solvents associated with metal parts cleaning, degreasing, painting, and paint cleanup; ferric chloride in printed circuit board etching; photo-developing solutions; cooling/cutting oils; heavy metal waste treatment sludge; plating/etching/stripping/plating line cleaning solutions; and laboratory packs and scrap metals. Gaseous substances include nitrogen, oxygen, carbon dioxide, argon, hydrogen, helium, acetylene, and other gas mixtures. The non-hazardous wastes that cannot be added to a dumpster or sewage line are related to packaging materials, tooling, printing, glass, and dirt [24].

### Stage Three—Flight Operations

European flight operations account for 3.8% of global $CO_2$ emissions contributing to effective radiative forcing. That leads to 4% of air flight-induced global warming through atmospheric interactions that can reinforce the warming impact. The oft-quoted figure is 81% generated by passenger flights and 19% by freight travel. While 60% of emissions are from international and 40% from domestic flights, 65% of its $CO_2$ emissions are in international airspace and therefore, are not necessarily associated with individual nation-states. Commercial aviation across Europe generated 707 million tons of $CO_2$ in 2013, which increased to 920 million tons in 2019, an increase of 30% in six years, estimated to triple by 2050 [23]. Long-haul flights (above 4000 km) represented approximately 6% of departures during 2019, causing 50% of all $CO_2$, NOx, sulfur oxides, unburnt hydrocarbon,

carbon monoxide, particulates, soot, and condensation trails. These emissions affect Earth's radiation balance and are a radioactive force from global aviation emissions contributing to climate change (Figure 2) [20,26].

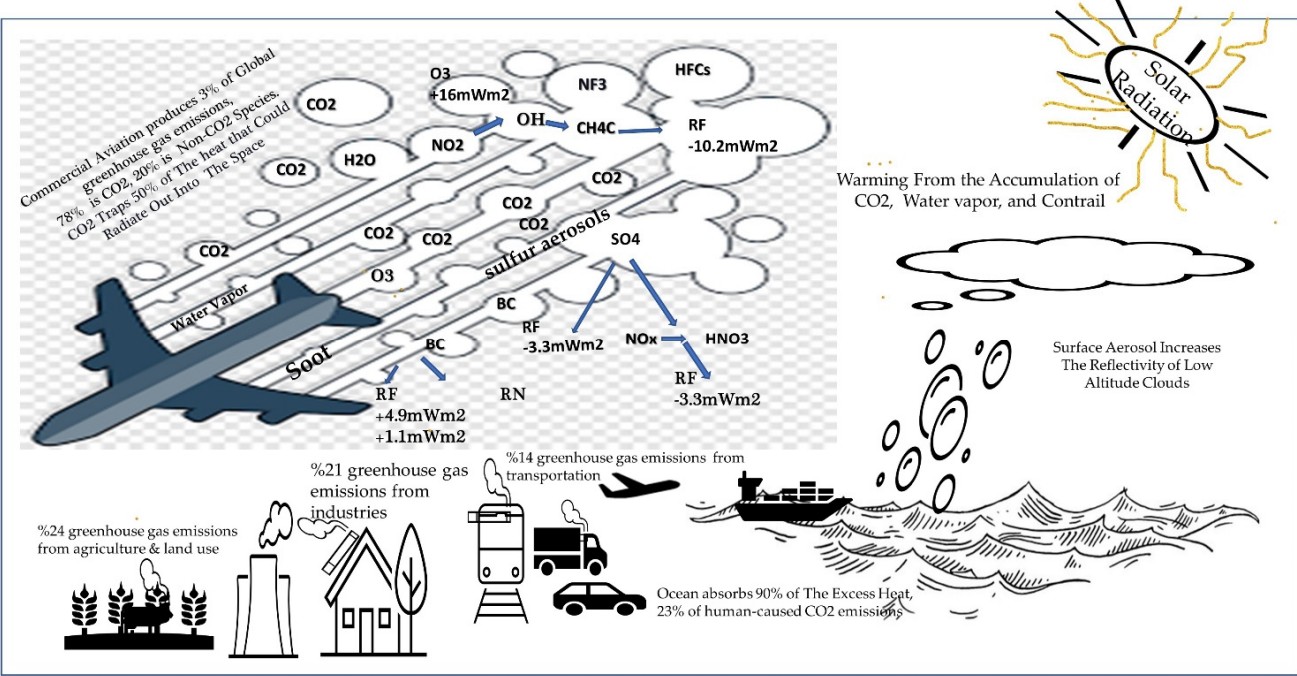

**Figure 2.** Climate forcing caused by greenhouse gas emissions from global aviation & other sources. Adopted from Nation Center for Environmental Information, assessing the global climate, January 2023 [19,26].

Comparing $CO_2$ emissions contributing to climate change by flights destination in 2019 (European and International flights), the North Atlantic (2.1% of flights) accounted for 14.5% of $CO_2$, Europe (4.5% of flights) 11.8% of $CO_2$, Asia Pacific (1.2% of flights) 13.7% of $CO_2$, Middle East (1.9% of flights) 7.2% $CO_2$, South Africa (0.5% of flights) 3.2% of $CO_2$, South Atlantic (0.4% of flights) 4.5% of $CO_2$, and Middle Atlantic (03% of flight) 3.1% of $CO_2$ [26,27].

$CO_2$ emissions of single-aisle aircraft were compared with twin-aisles to see the contributions to climate change in European and international flights; single-aisles have the larger share of flights. According to the European Aviation Environmental Report [23], 10% of $CO_2$ emissions from commercial aviation burning fossil fuel is during taxi, takeoff, initial climb up to cruise altitudes, and landing, and 90% is emitted 3000 to 40,000 feet above the ground. Even though international twin-aisle flights have a larger share of fuel burn, single-aisles within the EU27 and EFTA emit more $CO_2$, thus impacting climate change more than international flights [23]. It is estimated that between 2021 and 2050 approximately 21.2 gigatons of $CO_2$ will be released into the environment in a business-as-usual trajectory [27].

Further widespread problems are related to commercial flight noise and waste. The World Health Organization Europe [28] recommends noise levels below $L_{den}$ 45 dB (daily aircraft noise) and $L_{night}$ 40 dB (night aircraft noise) to prevent noise-induced health issues. Aircraft noise levels above 55 dB $L_{night}$ increase the risk of heart attacks and above 45 dB at $L_{night}$ increase the risk of hypertension, leading to hypertensive strokes and dementia. In 2019, 3.2 million people were exposed to noise levels above 55 dB $L_{den}$, and 1.3 million were exposed to above 50 $L_{night}$ in 98 major European airports [28]. These were 30% and 71% more than in 2005 [29]. The exposed population experienced annoyance with resentment,

displeasure, and discomfort; a decline in cognitive performance in school children; sleep disturbance; the development of cardiovascular disease; and hearing impairment.

Stage Three—Aviation Services

Sections related to commercial aviation services (Figure 3) generate an enormous quantity of solid and hazardous waste from passenger flights, flight catering centers, airport offices, retail outlets, restaurants, restrooms, air cargo terminals, ground services, landscaping, construction, and demolition projects [30]. In addition, airports generate water pollution due to their extensive handling of jet fuel and deicing chemicals.

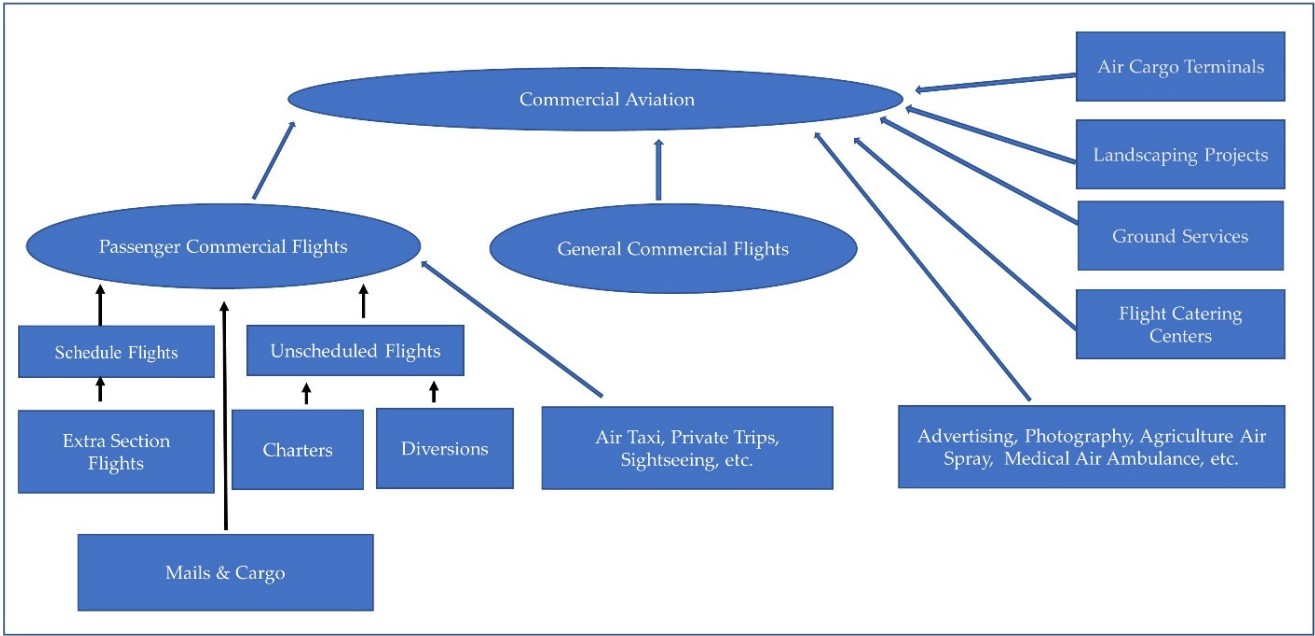

**Figure 3.** Sections related to commercial aviation services.

Between 2016 to 2022, for example, commercial passenger flights generated 5.2 million tons of waste, costing £400 million annually, most of which went to landfills or incineration. The Airports Council International report reveals that each passenger generates 0.82 kg to 2.5 kg of waste per flight, depending on distance and cabin class, with an average of 1.43 kg [31]. With an increasing number of air passengers from 798.6 million in 2008 to 1146.44 billion in 2019, European waste generated by European airports increased from 654.8766 million kg to 2866.1 billion kg in 2021 [32]. Of this waste, more than 35% consisted of cleaning wastes (such as paper, plastics, food, blankets, pillows, washroom bins, and medical waste) and catering waste (such as inflight meals, snacks, and beverages) destined for landfills or incinerators [2,31,33]. With the current growth in commercial aviation, the volume of waste generated will double by 2030 [8].

2.2.3. End-of-Service

Waste from end-of-service or retired aircraft has added challenges to controlling acid rain and air and water pollution. Between 1980 and 2015, with an average age of 25 to 28 years for passenger aircraft and 31 to 38 years for freighters, 15,534 commercial aircraft retired worldwide, with economic, operational, regulatory, safety, and environmental implications. With the growing number of aircraft in service, over the next ten years, 11,000 more aircraft will retire, while about 90% of the weight content of retired aircraft will be reused or recycled to save the Earth's resources [2].

The immediate challenge is that at an average age of twenty-four, seven hundred more jets are getting closer to the end of their lifespan each year, all containing hazardous materials that leak and contaminate the surrounding air, soil, and water. It is estimated that

18,000 will be decommissioned by 2030 [1]. This is an annual average of seven hundred aircraft in North America (38%), Europe (33%), Asia Pacific (11%), Latin America and the Caribbean (8%), Africa (7%), and the Middle East (3%). Additionally: (1) each aircraft has more than 350,000 valuable components, such as engines and electronics, with high economic value, potentially creating more than 30,000 tons of aluminum, 1800 tons of alloys, 1000 tons of carbon fiber, and 600 tons of other rare materials [4]. For example, the value of aircraft parts salvaged between 2009 and 2011 was more than USD2.5 billion. (2) The availability of critical metals are located in vulnerable economic and political regions. About 95% of Earth's supply of rare-earth metals come from China; (3) The higher global demand for rare natural resources is estimated to triple between 2018 and 2050. Other factors include (4) the increase in the world population; (5) rapid urbanization; (6) the rise in living standards; (7) changes in technological standards (new electronic innovations are leading to a 30 to 50% increase in demand for metals by 2030, and compared with 2010, the steel demand will grow by 90%; for copper, by 80%, for aluminum, nickel, and zinc, by 200%, while the gap in the copper supply may widen to 50%); (8) accessibility to hard-to-reach deposits and deep seabed mining; (9) there will be price volatility due to competition and the high cost of mining; and (10) energy demand is estimated to grow by 50% by 2030 [33,34].

In addition, decommissioning retired airplanes has caused pollution that has adversely affected human health, biodiversity, and climate change for the last century and continues to increase globally [24]. The effects related to the loss of ecosystems are the toxification of land, air, and water with poisonous pollutants that damage living organisms and their habitats. Industrial emissions of $CO_2$ are the main sources of greenhouse gases that contribute to the depletion of natural resources, often with irreversible consequences [35]. Thus, the transition to green aviation is valuable in moving toward green aviation.

## 3. Targets for Green Aviation

The National Aeronautics and Space Administration defines green aviation as the pursuit of reductions in noise, greenhouse gas emissions, and waste considered major commercial aviation pollution [36]. Sakar [37] stated that the fuel-efficiency of aircraft leads to operational efficiency and is a step toward green aviation, while Agarwal [38] considers airport infrastructure and the associated technology to facilitate green aviation. Marzova believes technology is important in green aviation, especially in new aircraft design [39]. Akhmatova et al. believe in total quality and supply chain management for transitioning into green aviation [40]. These two strategies have been in place in aerospace since the 20th century. Huete et al. recommend decentralizing civil aviation as an innovative phase to utilize hydrogen technology to help the industry achieve greater competitiveness for greenhouse reduction [41]. That jeopardizes coordinated communication, aviation rules, regulations, safety, security, and reliability in Europe and global air traffic.

In 2018 members of the Global Association of the Aviation Industry addressed green aviation by setting six targets: (1) fifty-five percent $CO_2$ emissions reduction of all flights n by 2030; (2) a reduction in net aviation $CO_2$ emissions by 69% by 2050, relative to 2005 levels, (3) reduce air flight's carbon footprint by 90% by 2050 compared to 1990; (4) thirty percent reduction in aircraft noise level by 2030 compared to 2017; (5) an annual average of 1.5% improvement in fuel efficiency; and (6) fifty-five percent reduction in premature deaths caused by air pollution by 2030 compared to 2005 [42].

In brief, the growth of commercial aviation across Europe contributes to greenhouse emissions affecting climate change through the aircraft lifecycle, including extraction of raw material, manufacturing, flight operation, ground services, and end-of-services. The challenges are that the demand for air traffic is expanding along with the size of the jet fuel market. It is also urgent that the sector's supply chain minimize greenhouse gas emissions below the 1990 level by 2030 and zero $CO_2$ emissions by 2050.

#### 4. Assessing the European Commercial Aviation Sector's Perceptions of Environmental Degradation

This study aimed to substantiate the impacts of the European commercial aviation sector's perceptions on environmental degradation during commercial flights, airport services, and aircraft life cycles. A study as such is pre-paradigmatic, needing a theoretical framework from which to make predictive hypotheses. There are, however, two propositions that can be made from the relationship between commercial aviation and environmental pollution:

Proposition one (P1). The European commercial aviation supply chain contributes to environmental degradation.

Proposition two (P2). Transformation into green aviation is a path to environmental sustainability.

To justify a proper methodology pertinent to the above propositions, we referred to Silverman's recommendation for qualitative research [43]. Seventeen online semi-structured interviews were conducted from September 2021 to December 2022 to draw on different perspectives of associates with entrenched interests in the region to address the two propositions. The propositions were used as open-ended questions that required detailed elaborations by the participants. The number of participants exceeded the twelve recommended by Adler and Adler [44] to attain information saturation in the inquiry process. Patterns were recurring with seventeen participants, and no added information emerged, as MacDougall and Fudge advised [45].

A signed agreement was extended to participants to maintain confidentiality. They were volunteers from airlines, aircraft manufacturers, and suppliers with knowledge of environmental, aerospace operations, and manufacturing dimensions associated with Air Malta, Malta Air, Air Albania, Aegean Airlines, Alitalia; The Civil Aviation Directorate within Transport Malta; The Malta Environment and Planning Authority; The European Regions Airline Association; European Environmental Agency; 3AC Composites; 3M Company; Phillips 66 Aviation, Airbus, and Boeing associates.

#### 5. Perspectives of Participants

Illuminating participants' perspectives regarding the two propositions, the following section is divided into perceptions of proposition one (P1) and proposition two (P2).

*5.1. Perceptions about Proposition One—The Commercial Aviation Supply Chain Contributes to Climate Change*

The interviewees unanimously approved P1 by stating that the European aviation sector creates 13.9% of transport emissions and impacts environmental pollution from aircraft operation generating greenhouse gases, noise, ground services, and airplane life cycle from creation to end-of-life decomposition. They added that the sector is only halfway toward cutting greenhouse gas emissions by 55% compared to the 1990 level by 2030. Thus, to achieve 2050 climate neutrality, European commercial aviation requires reducing gas emissions by 69% compared to 1990 levels.

*5.2. Perceptions about Proposition Two—Transformation into Green Aviation Is a Path to the Climate Pledge*

All interviewees supported proposition two. The participants explained that a combination of several actions will help the sector reduce greenhouse gases by a minimum of 69% to reach the 2030 and 2050 net-zero targets. Participants from Airbus and Boeing, major aircraft manufacturers, listed the integration of five modifications and their contributions to net-zero emissions, as presented in Figure 4. The factors are 1. airframe modifications (19%), 2. alternative propulsion systems (5%); 3. alternative aviation fuels (37%); 4. advancing air-traffic management (6%), and 5. advancing the manufacturing model to circular manufacturing (8%), [46].

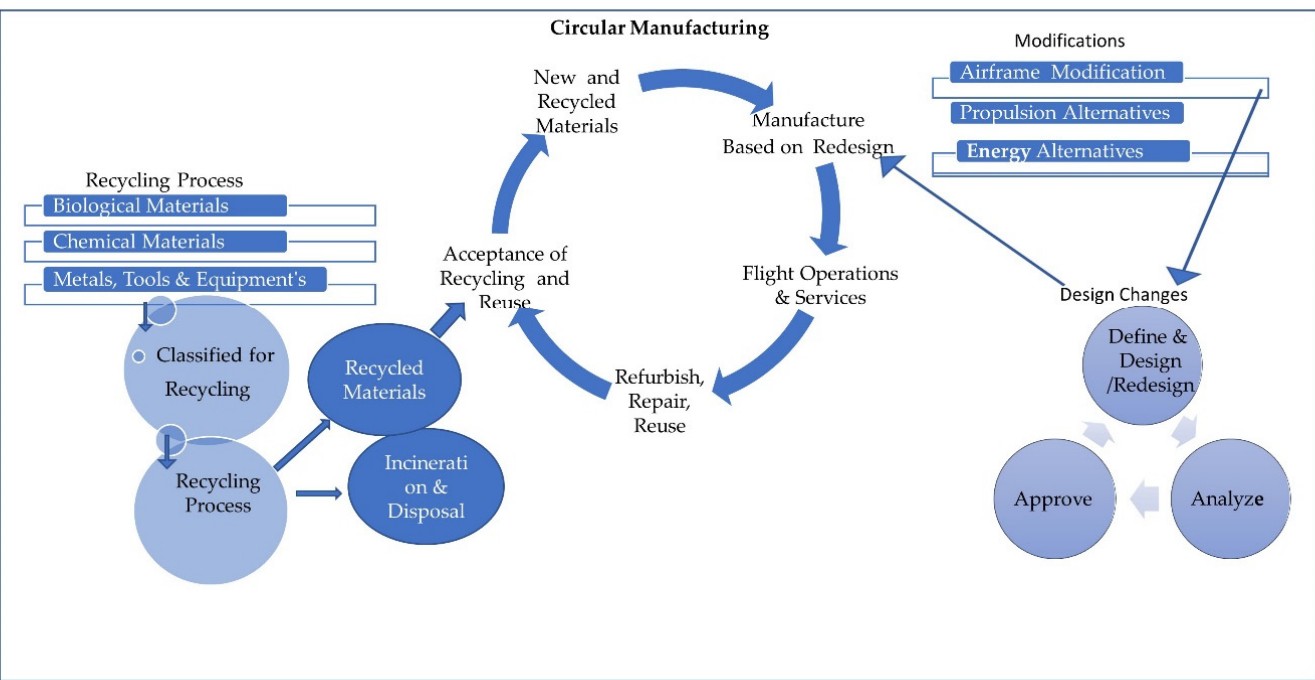

**Figure 4.** Aircraft modifications: Alternative energy, airframe modifications, & circular manufacturing model.

### 5.2.1. Airframe Modification

In the quest to reach zero $CO_2$ emission, Airbus is committed to airframe modification of new and derivative airplanes to maximize aircraft range and payload. Airbus and its supply chain have focused on aerodynamic efficiency. With induced drag being the dominant factor in aerodynamic efficiency, wings, surfaces, and lift devices remain priorities in manufacturing revolutionary airframe configurations implemented for 308 seats with overall dimensions and turnaround time similar to the 737-Boeing and 320-Airbus 320 aircraft. This aircraft operates with reduced fuel consumption, leading to 15% less per passenger-kilometer $CO_2$ and greenhouse emissions, 8% less noise, and a 5% increase in lift-to-drag ratio in cruise phases.

Lightweight, durable, and strong materials, such as carbon composites, are used for aerodynamic and fuel efficiency to reduce skin friction and lift drags while increasing effective wingspan in airframe and primary structures, such as Boeing 787, Boeing 777, Boeing 737, the Airbus 320, and Airbus 350. The materials have replaced 80% of aluminum parts in aircraft structures, such as engine blades, brackets, interiors, nacelles, propellers/rotors, single aisle wings, wide-body wings, and fuselage. The materials are also used to build aircraft components by the Airbus A380, A220, and A350X supply chain to deliver exceptional gains in weight and costs. Although the materials and structures technology have been developed and used in several new small and large aircraft, progress is still anticipated in acceptable margins linked to existing materials and innovative designs targeting improved assembly processes (e.g., bonding, stitching, and welding).

In addition, the on-ground folding wing-tip mechanism increases the effective in-flight wingspan and mitigates span constraints of existing airports' infrastructure. Wingspans fold upwards to reduce the wingspan to 212 feet and are compatible with parking gates. Structure multifunctionality and geometry adaptivity are other improvements for the structure to fulfill additional roles by modifying the material (for example, using nanotechnology).

### 5.2.2. Alternative Propulsion Systems

Additional aerodynamic efficiency is the propulsion system re-energizing the fuselage boundary layer to directly compensate for the viscous drag effects in the fuselage Wakefield

by fuselage wake-filling propulsion integration. The aircraft includes bio-inspired hybrid electroactive morphing and sensing for aerodynamical configuration, engaging intelligent electroactive actuators to modify the aircraft's lifting structure, and obtaining optimum aerodynamic performance.

Over the last decade, three technology paths have been used to reduce propulsion-system fuel consumption. They include increasing thermal efficiency by increasing the compressor overall pressure ratio with a consequent increase in core engine operating temperatures, increasing propulsive efficiency by increasing the engine bypass ratio and consequently, the fan diameter, and reducing installed engine weight and drag. The objectives are to develop and demonstrate breakthrough technologies for civil aircraft that could reduce $CO_2$ emissions by 20% (2025) to 30% (2035) at the aircraft level compared to current state-of-the-art aircraft.

### 5.2.3. Alternative Aviation Fuel

Further improvements are through alternative aviation fuel for environmental motives, operation costs, weight, storage capacity, wing size, and maximum load to carry for new aircraft designs from 2020 and newly built existing models from 2023. According to the Airbus and Boeing participants, each new generation of aircraft (such as ATR-600, Embraer E2, Bombardier C-series, Boeing 787, and Airbus 350) produces 80% less $CO_2$ per seat than the first jets in the 1950s. Especially A350 and 737MAX consume less than three liters per one hundred passenger kilometers. That is comparable to the fuel consumption of a compact car. In 2022 Airbus commercial aircraft and helicopters will be certified to fly with up to a 50% blend of sustainable aviation fuel to achieve certification of 100% capability by 2030 for commercial and military aircraft and helicopters. Sustainable aviation fuel supply is expected to account for 83% of the total fuel consumption by 2050.

Electric energy is the second method at the early research and development stage to reduce ground traffic congestion and $CO_2$ emissions. Suppliers are working to increase battery energy density storage for hybrid-electric and fully electric aircraft used for training flights and two-to-four-person 'air taxi' (commuter flights) operations by 2023–2024. Suppliers, such as Zunum and Wright Electric, are developing more commercial, regional short-haul aircraft by 2035. Aircraft powered by electricity, hydrogen, and solar technology are steps toward a climate neutral. The purpose is to enable the aviation industry to reduce $CO_2$ emissions drastically. Aircraft and engine technology improvements are expected to include hydrogen-powered aircraft on intra-EU routes from 2035 onwards, a step-change in energy efficiency from new aircraft types in the next ten years and an optimized range and capacity of hybrid-electric aircraft. This will require high technology readiness by 2027–2030, new certification procedures for disruptive technologies, and accelerated fleet renewal. A hybrid combines liquid aviation fuel with electric propulsion for mid- to long-range flights for efficiency and environmentally sustainable performance.

Biogas is the third option that is the main product of the anaerobic digestion of wet biomass, primarily gases, including methane and carbon dioxide, with minimum sulfide, moisture, and siloxanes. The gases can be oxidized with oxygen-releasing energy to convert into energy and renewable bio-fertilizer. Alternatively, biogas can be upgraded to bio-methane (CH4) to replace natural gas or be fed into the existing gas infrastructure to reduce fossil fuel consumption and decarbonize toward net zero emissions. Biogas and biomethane contribute to energy security since the production system can be established and operated locally, prevent water pollution, minimize the pathogen content of slurries, produce bio-fertilizers, and improve air quality by reducing fugitive emissions.

### 5.2.4. Improvements in Air Traffic Management

Air traffic management is a ground-based service to control traffic in and around airports, airport terminals, and airspace and provide advisory information to aircraft in non-controlled airspace. According to Air Malta, Malta Air, Air Albania, Aegean Airlines, Alitalia, and The Civil Aviation Directorate within Transport Malta, improvements in air

traffic management are an effective way to reach the 2050 net-zero emission by implementing the Single European Sky by 2020. This is a step towards a network-centric and digital air traffic management system, accelerated innovation, and rapid decarbonization of ground operations by 6–9%, being in a priority pending proposal status.

Participants believe that alternative aviation fuel production and deployment must be scaled up to make this a reality, ensuring robust and transparent sustainability criteria and a diversified and sustainable feedstock base. An increase in the maximum blending ratio from 50 to 100% is also required. Finally, market-based measures are crucial in the short term. By 2030, market-based measures could be responsible for 27% $CO_2$ reductions. By 2050, as the sector relies more on in-sector reductions, the maximum blending ratio could be responsible for 80% $CO_2$ reductions. In the meantime, using environmentally friendly materials is the short-term and long-term strategy for changing the manufacturing model.

### 5.2.5. Manufacturing Models

Airbus commercial aircraft manufacturing entails the direction and velocity of technological trends associated with Industry 4.0 and 5.0, aiming to achieve its greenhouse gas reduction goal by 2030 and net-zero emission by 2050. The manufacturing revolutionary paths, between Industry 1.0 to 3.0 from the 18th to 20th century, negatively impacted environmental sustainability by increasing productivity in pursuit of wealth. Thus, expanding urbanization, overcrowding cities by attracting laborers, and generating excessive industrial waste and pollution became the norm. In addition, the Industrial Revolutions 1.0 through 3.0 rippled throughout the Earth's ecological spheres, exhausting the planet's stock of valuable natural capital, such as water, trees, soil, rocks and minerals, wild and domesticated animals, etc. That led to global water and air pollution challenges, biodiversity reduction, and wildlife habitat and land destruction.

In the 21st century, Industry 4.0 was followed by the 5.0 Revolution, as the entire realm of aerospace manufacturing was founded on advanced technologies. Industry 4.0 and Industry 5.0 integrate the industrial Internet of Things (IoT), cloud computing and analytics, 3D, blockchain, artificial intelligence, and machine learning into the operating system, putting Airbus on the path toward full digital transformation across the entire aircraft lifecycle.

The manufacturing model is based on a digital circular economy with reduction, reuse, recycling, and reproduction, an alternative to a linear model. The linear model focuses on profitability, irrespective of the product life cycle, whereas the circular model targets sustainability. The circular model is a closed-loop system that eradicates waste by recovering materials through a calculated reengineering process to modify product components. At the same time, the prevailing linear model leads to depleting finite Earth resources and accumulating wastes [35]. Circularity aims to use renewable energy sources and eradicate waste throughout a calculated process from design to production and consumption [39,47].

While the use of Earth's resources is optimized, and recycled materials become integrated into products while the characteristics of the final products satisfy the desired values, business process reengineering stands at the forefront of eco-design. The business process requires longer product life cycles, using renewable energy, minimizing the use of natural materials by recycling Earth's resources in value chains and eliminating hazardous and non-hazardous materials.

In 2020 almost 80% of the global production process was dedicated to using 3D or additive manufacturing to produce aircraft and spare parts. For example, General Electric and Rolls-Royce, Boeing, and Airbus suppliers have invested in 3D manufacturing of small parts, such as jet-fuel nozzles, for flexible design, rapid prototyping, print on demand, strong and lightweight parts, and fast design and production to minimize waste and use of resources. Using laser sets, parts are made by layers from metal powder or composite materials, deposited, joined, or solidified under computer control [43].

Further use of technology is blockchain transaction that contributes to environmental sustainability. Blockchain has enabled ledgers to be shared, updated, validated, and shared across participants, creating secure and supportable transactions without mediators or central databases. Adopting blockchain would enable companies to reduce resources and time in a wide range of repetitious procedures, such as billing between multiple airlines and travel agencies, settling loyalty points or rewards, purchasing travel insurance, and paying airport fees.

## 6. Conclusions

Commercial aviation growth has been regarded as an important contributor to the mobility and economy of Europe. Along with the growth, environmental degradation caused by the aviation supply chain generating pollution and emissions increased. Between 2005 and pre-pandemic 2020, the $CO_2$ emitted by commercial aviation-passenger and cargo increased by 50%. Although with the 2020 commercial aviation paralysis, $CO_2$ emissions declined by more than half but escalated to 86% of the pre-pandemic in 2022. In addition, noise, overconsumption, and pollution of Earth's resources, including materials, water, energy, and fertile land, have rapidly increased, causing irreversible environmental damage.

This study examines a European path to environmental sustainability toward a green commercial aviation supply chain. Driven by literature and a review of related documents, two propositions were advanced to orient perspectives on the relationship between commercial aviation supply chains and environmental pollution. Seventeen online semi-structured interviews were conducted to address the two propositions as open-ended questions. They include: proposition one—European commercial aviation supply chain contributes to environmental degradation and proposition two—transformation into green aviation is a path to environmental sustainability. The participants unanimously approved proposition one by endorsing pollution sources in the commercial aviation supply chain during four stages of an aircraft life cycle, from mining and processing raw materials to aircraft manufacturing, flight operations, ground services, and aircraft decommissioning, each of which underlines the overconsumption of natural resources and environmental pollution. They expressed that consuming fossil fuels generates greenhouse gas emissions and increases $CO_2$ concentration in the atmosphere. Therefore, the pledge to net-zero emissions by 2050 requires transforming into green commercial aviation through widespread deployments of innovative technologies, i.e., proposition two.

The aviation supply chain, especially the original aircraft manufacturers, Airbus and Boeing, has focused on a combination of several actions to reduce greenhouse gases. The factors include airframe modifications, alternative propulsion systems, alternative aviation fuels, advancing air-traffic management, and shifting the manufacturing process from the linear model (extract-manufacture, operate, dispose of) to a digital circular model (a recycling system for maximizing efficiency and reducing waste). These six factors for new and derivative airplanes present a significant reduction in fossil fuel burning, leading to negligible/zero carbon emissions to achieve 2030 greenhouse gas emission reductions and zero $CO_2$ emission by 2050.

**Author Contributions:** Methodology, B.M.F.; Software, A.A. (Alexander Ansari); Validation, A.A. (Al Ansari); Formal analysis, A.A. (Alexander Ansari); Investigation, B.M.F. and A.A. (Al Ansari); Resources, A.A. (Al Ansari); Data curation, A.A. (Alexander Ansari); Writing—original draft, B.M.F.; Writing—review & editing, A.A. (Alexander Ansari). All authors have read and agreed to the published version of the manuscript.

**Funding:** This research received no external funding.

**Institutional Review Board Statement:** No applicable.

**Informed Consent Statement:** No applicable.

**Data Availability Statement:** Published the results not raw data.

**Conflicts of Interest:** The authors declare no conflict of interest.

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
