# Peer review of "Green Commercial Aviation Supply Chain—A European Path to Environmental Sustainability"

_sustainability, doi:10.3390/su15086574_

Round 1

Reviewer 1 Report

Dear Authors,

The manuscript sustainability-2276247, entitled "Green Commercial Aviation Supply Chain - A European Path to Environmental Sustainability" deals with an interesting and relevant topic within the field of prospects for sustainability. However, to be considered a Scientific Article, it needs better planning and organization of Materials and Methods, as it is not clear how the interviews were conducted. In addition, the results achieved are not clearly presented. Even understanding that this is a qualitative study, the way in which the results are presented becomes confusing to the reader. At the same time, the discussions were made too simple. Therefore, for this study to be published as an article, the Material and Methods, Results and Discussion sections need to be completely redone.

In view of the topics addressed in this study, I would recommend that changes be made to convert it into a review study, with all the necessary adjustments being made.

My assessment is that major revision is needed.

The specific comments below should also be heeded.

Specific comments:

- Reviewer (Lines 5-6): Indicate complete affiliation information, with Institution, Address and e-mail.

- Reviewer (Lines 12-14): The objective of the study is only implied in this excerpt, and it is not clear to the reader. Please rewrite, highlighting the objective clearly and according to the scientific method.

- Reviewer (Line 20): The expression "Commercial aviation" is repeated in the title and keywords. Please replace by adding another suitable expression.

- Reviewer (Lines 22-292): The manuscript does not follow the journal's standard formatting, in which it is recommended to use a 4.6 cm indent to the left. Please adapt the main document to the Sustainability rules.

- Reviewer (Line 30): Include "respectively", after "by 75.6% and 56%".

- Reviewer (Lines 41-42): The CO2 is also a greenhouse gas. Therefore, it should not be quoted separately. Rewrite this passage, leaving only "...their commercial aviation must curb greenhouse emissions".

- Reviewer (Lines 48-49): The objective of the manuscript must be rewritten, leaving it in line with the scientific method. From the way the manuscript was organized, my understanding is that it is a review, and this should be clear in the objective.

- Reviewer (Line 49-50): Replace "The following section..." with "This review study..."

- Reviewer (Lines 50-52): Please rewrite the snippet. Replace "First is a literature review on " by "The first section addresses".

- Reviewer (Line 52): Please rewrite the snippet. Replace "Second is the list..." with "The second section describes a list...".

- Reviewer (Lines 52-53): Please rewrite the snippet. Replace "The third is the..." by "The third presents a...".

- Reviewer (Line 53): Please rewrite the snippet. Replace "Fourth is the qualitative research findings. " by "The fourth brings qualitative results obtained through the application of the methodology presented in the third section".

- Reviewer (Line 54): As I understand it, this section should be titled "Relationship between Aviation and Environmental Pollution". Please review this.

- Reviewer (Line 56): Include "European Environmental Agency" before [10].

- Reviewer (Lines 61-62): Delete "[1] International Civil Aviation 61 Organization [14]" and add "[1,14]".

- Reviewer (Line 68): Did you mean "Zettajoules"? Please correct this error.

- Reviewer (Line 105): Regarding the title of the figure, which has not yet been added, it must be added to the body of the main document, below the figure, and be self-explanatory. Regarding the figure itself, it would be interesting to change the font of the texts, placing Palatino Linotype, in size 10 or 8, to standardize with the body of the text. The image font must follow the Sustainability citation guidelines, and be placed at the end of the title.

- Reviewer (Lines 108-111): Where was this information obtained from? No source was reported. Please quote.

- Reviewer (Line 131): The figure title is not self-explanatory and its position does not follow the Sustainability Author Guidelines. In addition, picture text should be in the same font and size as the body text of the document. The map and text at the bottom of the figure are not understandable. Please correct it.

- Reviewer (Line 138): Replace "Of which" with "From these waste".

- Reviewer (Lines 150-151): And for Europe, what is the estimate? It would be interesting to add this information.

- Reviewer (Line 158): The title of the figure should be improved, and changed to below the figure.

- Reviewer (Lines 183-191): Add "1)" to the first item and describe more clearly what each item refers to. The organization of these topics is not clear, it needs to be revised and organized.

- Reviewer (Line 193): What did you mean by the term "These stresses"? Make a connection more clearly with the previous paragraph.

- Reviewer (Lines 204-215): This section has been covered very superficially and needs to be improved. More references need to be cited and goals for green aviation discussed.

- Reviewer (Line 207-211): Include ":" before "1)" and standardize the use of ";" after each topic.

- Reviewer (Line 211): Include "and" before "6)".

- Reviewer (Line 215): As already mentioned, due to the methodology used and the results generated, my understanding is that this study cannot be characterized as a scientific article. I consider that the methodology is too simple, and the results are not very conclusive. For this reason, my recommendation is that it be converted into a review article, and that this section be titled "Assessing the European commercial aviation sector's perceptions of environmental degradation".

- Reviewer (Lines 218-219): Replace "There are, however, two propositions that can be advanced from the literature review:" with "Two propositions can be made from the relationship between Aviation and Environmental Pollution".

- Reviewer (Lines 224-228): The questions that made up the questionnaire used were not presented. Which was? Was it just two questions, as implied in the later section? Were the answers free? Could this qualitative questionnaire have its answers converted into numerical ones, for the generation of graphs? The description of the methodology is too simple and does not allow understanding how the study was carried out. Only the application of two questions makes the scientific article very simplistic. Please review this.

- Reviewer (Line 234): As I understand it, this section should be titled "Perceptions of the European Commercial Aviation Industry on Environmental Degradation". Considering this study as a "Review", the perceptions of the participants of the aviation companies can be maintained, if they are duly cited. I recommend you do this.

- Reviewer (Line 237): The subsection title should be self-explanatory. Replace "Discussion on P1" with " Perceptions about the Preposition One".

- Reviewer (Line 237): The body text of the subsection must start in a new paragraph. Please do that and start the paragraph on the next line.

- Reviewer (Line 243): The subsection title should be self-explanatory. Replace "Discussion on P2" with " Perceptions about the Preposition Two".

- Reviewer (Lines 243): The body text of the subsection must start in a new paragraph. Please do that and start the paragraph on the next line.

- Reviewer (Line 248): Replace "presented" with "illustrated".

- Reviewer (Line 250): The Figure must be adjusted to the rules of the journal.

- Reviewer (Line 371): The- Reviewer (Line 234): As I understand it, this section should be titled "Final considerations", to close the manuscript.

- Reviewer (Line 392): Author Contributions, Funding, Institutional Review Board Statement, Informed Consent Statement, Data Availability Statement, Acknowledgments and Conflicts of Interest, as required by Sustainability, were not entered. Please do this.

- Reviewer (Line 393): The references section does not comply with the Sustainability Instructions for Authors and must be fully adequate.

Best regards,

Reviewer

Reviewer 2 Report

This is an interesting paper summarizing what goes on in aviation regarding green transition.

Regarding changes, I would like see P1 and P2 more clearly stated as conclusion in the summary since these are the focus on methodology and the focus coming out of the introduction. That would wrap up the whole paper better.

The main issue is how the aviation industry can become more environmental sustainable

This is relevant for sustainability since they first describe the problem and then discuss possible contributions to sustainability.

To see it all collected together for sustainability is original in my view even if most (or possible all) parts can be found in other publications.

The contribution is to present all these contributions to sustainability together to make an overview of the issue for aviation industry

This paper is well written.

The conclusions are consistent with the evidence and arguments presented, but as you saw from my comment in the review I wished for the P1 and P2 to become more visible in the summary as they now tends to merge into a more general discussion. There are no explicit conclusion in the paper, but in my view that is not a problem since the paper presents many possible contributions for the industry to become environmental sustainable and this is an ongoing process more than a final goal.

Reviewer 3 Report

This paper describes the impact of European commercial aviation on climate change based on the literature survey where two propositions are presented to decrease the pollution generated by the European commercial aviation supply chain through commercial flights, airport operations, and the airplane life cycle stages. The paper presents a multispectral perspective by deploying innovative technologies to modify airframes, aviation fuel composition, utilizing alternative propulsion systems, adjusting manufacturing models, and improving air traffic management. An improved bio-feul composition can be effective in a significant reduction in fossil fuel burning, leading to negligible / zero carbon emissions to achieve 2050 targets.

The paper is well written and the arguments are strengthened by data. However, comparative tables are missing for the analytical part that can better represent the gaps.

Also include the Green supply chain management (GSCM) and Total Quality Management (TQM) for reducing environmental risks in aviation. 

Some excellent references are missing in the paper. For example:

J. Huete, D. Nalianda, B. Zaghari and P. Pilidis, "A Strategy to Decarbonize Civil Aviation: A phased innovation approach to hydrogen technologies," in IEEE Electrification Magazine, vol. 10, no. 2, pp. 27-33, June 2022, doi: 10.1109/MELE.2022.3166245.

Malika-Sofi Akhmatova, Antonina Deniskina, Dzhennet-Mari Akhmatova, Anna Kapustkina, Green SCM and TQM for reducing environmental impacts and enhancing performance in the aviation spares supply chain, Transportation Research Procedia, Volume 63, 2022, Pages 1505-1511, ISSN 2352-1465, https://doi.org/10.1016/j.trpro.2022.06.162.

Round 2

Reviewer 1 Report

- Reviewer (1st review):

The manuscript sustainability-2276247, entitled "Green Commercial Aviation Supply Chain - A European Path to Environmental Sustainability" deals with an interesting and relevant topic within the field of prospects for sustainability. However, to be considered a Scientific Article, it needs better planning and organization of Materials and Methods, as it is not clear how the interviews were conducted. In addition, the results achieved are not clearly presented. Even understanding that this is a qualitative study, the way in which the results are presented becomes confusing to the reader. At the same time, the discussions were made too simple. Therefore, for this study to be published as an article, the Material and Methods, Results and Discussion sections need to be completely redone.

In view of the topics addressed in this study, I would recommend that changes be made to convert it into a review study, with all the necessary adjustments being made.

My assessment is that major revision is needed.

The specific comments below should also be heeded.

- Authors:

All Comments were corrected, except the comment on Lines 41-42 was refuted because greenhouse gas and CO2 are not synonymous. Greenhouse gases are 79% CO2 and non-CO2 gases.

- Reviewer (2nd review):

Following what we mentioned in the first review, the manuscript deals with an interesting and relevant topic within the field of perspectives for sustainability and has scientific merit.

Most of the recommendations were followed, and I understand that it was potentially improved. In addition, I reinforce the need to prepare figures with better graphic quality, and I recommend that you consider applying the specific comments that were not considered (specified below).

My assessment is that minor revisions are still needed.

Specific comments:

- Reviewer (Lines 5-6): Indicate complete affiliation information, with Institution, Address and e-mail.

  Authors: It is done.

  Reviewer (2nd review): This was not done. In the document indicated, only the institution is indicated, without the address, as well as the e-mails of the authors. Please correct.

- Reviewer (Lines 12-14): The objective of the study is only implied in this excerpt, and it is not clear to the reader. Please rewrite, highlighting the objective clearly and according to the scientific method.

  Authors: Done.

  Reviewer (2nd review): Ok.

- Reviewer (Line 20): The expression "Commercial aviation" is repeated in the title and keywords. Please replace by adding another suitable expression.

  Authors: Done.

  Reviewer (2nd review): Ok.

- Reviewer (Lines 22-292): The manuscript does not follow the journal's standard formatting, in which it is recommended to use a 4.6 cm indent to the left. Please adapt the main document to the Sustainability rules.

  Authors: Below is a copy of the Journal’s Free Format Submission. 

“Free Format Submission”

“Sustainability now accepts free format submission:

•             We do not have strict formatting requirements, but all manuscripts must contain the required sections: Author Information, Abstract, Keywords, Introduction, Materials & Methods, Results, Conclusions, Figures and Tables with Captions, Funding Information, Author Contributions, Conflict of Interest and other Ethics Statements. Check the Journal Instructions for Authors for more details.

•             Your references may be in any style, provided that you use the consistent formatting throughout. It is essential to include author(s) name(s), journal or book title, article or chapter title (where required), year of publication, volume and issue (where appropriate) and pagination. DOI numbers (Digital Object Identifier) are not mandatory but highly encouraged. The bibliography software package EndNote, Zotero, Mendeley, Reference Manager are recommended.

•             When your manuscript reaches the revision stage, you will be requested to format the manuscript according to the journal guidelines.”

  Reviewer (2nd review): OK. Although they accept it in free format, I recommend that they already format it in accordance with the guidelines for authors, so that it is not necessary in the following steps. It's good practice.

- Reviewer (Line 30): Include "respectively", after "by 75.6% and 56%".

  Authors: Done,

  Reviewer (2nd review): Ok.

- Reviewer (Lines 41-42): The CO2 is also a greenhouse gas. Therefore, it should not be quoted separately. Rewrite this passage, leaving only "...their commercial aviation must curb greenhouse emissions".

  Authors: Done. I believe 79% of greenhouse gas emission is CO2.  The CO2 emission is also from other sources. Therefore, it has to be expressed separately. The immediate target is to reduce greenhouse gas emissions by 55% from 2010 levels by 2030 compared to the 1990 level and zero CO2 emissions by 2050.

  Reviewer (2nd review): OK. Although I do not fully agree, I understand that it is well written and that it can remain this way, being the authors' option.

- Reviewer (Lines 48-49): The objective of the manuscript must be rewritten, leaving it in line with the scientific method. From the way the manuscript was organized, my understanding is that it is a review, and this should be clear in the objective.

  Authors: Done.

  Reviewer (2nd review): Ok.

- Reviewer (Line 49-50): Replace "The following section..." with "This review study..."

  Authors: Done.

  Reviewer (2nd review): Ok.

- Reviewer (Lines 50-52): Please rewrite the snippet. Replace "First is a literature review on " by "The first section addresses".

  Authors: Done.

  Reviewer (2nd review): Ok.

- Reviewer (Line 52): Please rewrite the snippet. Replace "Second is the list..." with "The second section describes a list...".

  Authors: Done.

  Reviewer (2nd review): Ok.

- Reviewer (Lines 52-53): Please rewrite the snippet. Replace "The third is the..." by "The third presents a...".

  Authors: Done.

  Reviewer (2nd review): Ok.

- Reviewer (Line 53): Please rewrite the snippet. Replace "Fourth is the qualitative research findings. " by "The fourth brings qualitative results obtained through the application of the methodology presented in the third section".

  Authors: Done.

  Reviewer (2nd review): Ok.

- Reviewer (Line 54): As I understand it, this section should be titled "Relationship between Aviation and Environmental Pollution". Please review this.

  Authors: Done. This section is organized and numbered based on Figure 1-aircraft lifecycle, for the flow of the information and prevents the confusion mentioned by the reviewer.

  Reviewer (2nd review): Ok.

- Reviewer (Line 56): Include "European Environmental Agency" before [10].

  Authors: Done.

  Reviewer (2nd review): The citation number should follow the authors, as follows: "According to the European Environmental Agency [10], climate..."

- Reviewer (Lines 61-62): Delete "[1] International Civil Aviation 61 Organization [14]" and add "[1,14]".

  Authors: Done.

  Reviewer (2nd review): Ok.

- Reviewer (Line 68): Did you mean "Zettajoules"? Please correct this error.

  Authors: Done.

  Reviewer (2nd review): Ok.

- Reviewer (Line 105): Regarding the title of the figure, which has not yet been added, it must be added to the body of the main document, below the figure, and be self-explanatory. Regarding the figure itself, it would be interesting to change the font of the texts, placing Palatino Linotype, in size 10 or 8, to standardize with the body of the text. The image font must follow the Sustainability citation guidelines, and be placed at the end of the title.

  Authors: Done.

  Reviewer (2nd review): In the revised version, the figure is on line 152. A title has been included, but the graphic quality of the figure has not been improved. I strongly recommend that you improve the graphic quality of the figure.

- Reviewer (Lines 108-111): Where was this information obtained from? No source was reported. Please quote.

  Authors: Done.

  Reviewer (2nd review): This passage is between lines 147-150 in the revised document, but no reference to the data has been included. Why? Please fix this.

- Reviewer (Line 131): The figure title is not self-explanatory and its position does not follow the Sustainability Author Guidelines. In addition, picture text should be in the same font and size as the body text of the document. The map and text at the bottom of the figure are not understandable. Please correct it.

  Authors: Done.

  Reviewer (2nd review): It was corrected in parts, but the graphic quality could be better (see figure 1). Please, improve the graphic quality of the figure, if possible.

- Reviewer (Line 138): Replace "Of which" with "From these waste".

  Authors: Done.

  Reviewer (2nd review): Ok.

- Reviewer (Lines 150-151): And for Europe, what is the estimate? It would be interesting to add this information.

  Authors: Done.

  Reviewer (2nd review): Ok.

- Reviewer (Line 158): The title of the figure should be improved, and changed to below the figure.

  Authors: Ok.

  Reviewer (2nd review): Figure 3 (1 in the revised document) had its graphical quality potentially improved. The same should have been done with the others.

- Reviewer (Lines 183-191): Add "1)" to the first item and describe more clearly what each item refers to. The organization of these topics is not clear, it needs to be revised and organized.

  Authors: Done. The numbering does apply.

  Reviewer (2nd review): Ok.

- Reviewer (Line 193): What did you mean by the term "These stresses"? Make a connection more clearly with the previous paragraph.

  Authors: Done.

  Reviewer (2nd review): Ok.

- Reviewer (Lines 204-215): This section has been covered very superficially and needs to be improved. More references need to be cited and goals for green aviation discussed.

  Authors: Done. This section was improved, and more information on green aviation was obtained in interviews.

  Reviewer (2nd review): Ok.

- Reviewer (Line 207-211): Include ":" before "1)" and standardize the use of ";" after each topic.

  Authors: Done.

  Reviewer (2nd review): Ok.

- Reviewer (Line 211): Include "and" before "6)".

  Authors: Done.

  Reviewer (2nd review): Ok.

- Reviewer (Line 215): As already mentioned, due to the methodology used and the results generated, my understanding is that this study cannot be characterized as a scientific article. I consider that the methodology is too simple, and the results are not very conclusive. For this reason, my recommendation is that it be converted into a review article, and that this section be titled "Assessing the European commercial aviation sector's perceptions of environmental degradation".

  Authors: Done.

  Reviewer (2nd review): Ok.

- Reviewer (Lines 218-219): Replace "There are, however, two propositions that can be advanced from the literature review:" with "Two propositions can be made from the relationship between Aviation and Environmental Pollution".

  Authors:

  Reviewer (2nd review): This recommendation was not followed. Please check and correct.

- Reviewer (Lines 224-228): The questions that made up the questionnaire used were not presented. Which was? Was it just two questions, as implied in the later section? Were the answers free? Could this qualitative questionnaire have its answers converted into numerical ones, for the generation of graphs? The description of the methodology is too simple and does not allow understanding how the study was carried out. Only the application of two questions makes the scientific article very simplistic. Please review this.

  Authors: There are two propositions, not a questionnaire, that is tested in open-ended interviews. So, based on qualitative discussions, I can only present their comment. There is no data at his time to test the airlines and aircraft manufacturers since implementing new technologies and data collection after implementation on their developments may take a decade.

  Reviewer (2nd review): OK. This information should be clearer to the reader. I suggest you do this in the appropriate sections.

- Reviewer (Line 234): As I understand it, this section should be titled "Perceptions of the European Commercial Aviation Industry on Environmental Degradation". Considering this study as a "Review", the perceptions of the participants of the aviation companies can be maintained, if they are duly cited. I recommend you do this.

  Authors: Done.

  Reviewer (2nd review): Ok.

- Reviewer (Line 237): The subsection title should be self-explanatory. Replace "Discussion on P1" with " Perceptions about the Preposition One".

  Authors: Done.

  Reviewer (2nd review): Ok.

- Reviewer (Line 237): The body text of the subsection must start in a new paragraph. Please do that and start the paragraph on the next line.

  Authors: Done.

  Reviewer (2nd review): Ok.

- Reviewer (Line 243): The subsection title should be self-explanatory. Replace "Discussion on P2" with " Perceptions about the Preposition Two".

  Authors: Done.

  Reviewer (2nd review): Ok.

- Reviewer (Lines 243): The body text of the subsection must start in a new paragraph. Please do that and start the paragraph on the next line.

  Authors: Done.

  Reviewer (2nd review): Ok.

- Reviewer (Line 248): Replace "presented" with "illustrated".

  Authors: Done.

  Reviewer (2nd review): Ok.

- Reviewer (Line 250): The Figure must be adjusted to the rules of the journal.

  Authors: Done.

  Reviewer (2nd review): Figure has been improved, but its graphic quality is still low. I insist on recommending that you improve all the figures, changing the font and size of the letters, as well as putting images with better resolution.

- Reviewer (Line 371): As I understand it, this section should be titled "Final considerations", to close the manuscript.

  Authors: The last part is the Summary, not the Final Findings, to follow the journal format.

  Reviewer (2nd review): If the authors choose to keep this form, and the format allows it, I agree.

- Revisor (Linha 392): Contribuições do Autor, Financiamento, Declaração do Conselho de Revisão Institucional, Declaração de Consentimento Livre e Esclarecido, Declaração de Disponibilidade de Dados, Agradecimentos e Conflitos de Interesse, conforme exigido pela Sustentabilidade, não foram inseridos. Por favor faça isso.

  Autores:

  Revisor (2ª revisão ): A recomendação não foi seguida. Embora o Sustainability aceite em formato livre, é uma boa prática enviar manuscritos devidamente formatados. Facilita a etapa de edição.

- Revisor (Linha 393): A seção de referências não está de acordo com as Instruções de Sustentabilidade para Autores e deve ser totalmente adequada.

  Autores:

  Revisor (2ª revisão ): Igual ao comentário anterior.

Atenciosamente,

Revisor

Reviewer 3 Report

The paper has significantly improved after incorporating the imporant comments of the reviewers in the first round. I see no further corrections in the paper.

Author Response

Reviewer # 3 made no comment for round 2.